# Analysis of Ice Phenology of Middle and Large Lakes on the Tibetan Plateau

**DOI:** 10.3390/s23031661

**Published:** 2023-02-02

**Authors:** Lijun Sun, Binbin Wang, Yaoming Ma, Xingdong Shi, Yan Wang

**Affiliations:** 1Land-Atmospheric Interaction and Its Climatic Effects Group, State Key Laboratory of Tibetan Plateau Earth System, Environment and Resources (TPESER), Institute of Tibetan Plateau Research, Chinese Academy of Sciences, Beijing 100101, China; 2College of Earth and Planetary Sciences, University of Chinese Academy of Sciences, Beijing 100049, China; 3National Observation and Research Station for Qomolongma Special Atmospheric Processes and Environmental Changes, Shigatse 858200, China; 4China-Pakistan Joint Research Center on Earth Sciences, Chinese Academy of Sciences, Islamabad 45320, Pakistan; 5College of Atmospheric Science, Lanzhou University, Lanzhou 730000, China; 6Kathmandu Center of Research and Education, Chinese Academy of Sciences, Beijing 100101, China; 7Key Laboratory of Land Surface Pattern and Simulation, Institute of Geographic Science and Natural Resources Research, Chinese Academy of Sciences, Beijing 100101, China

**Keywords:** Tibetan Plateau, lake ice phenology, passive microwave, MODIS, climate change

## Abstract

Considered as a sensitive indicator of climate change, lake ice phenology can have significant influences on regional climate by affecting lake-atmosphere energy and water exchange. However, in situ measurements of ice phenology events are quite limited over high-elevation lakes on the Tibetan Plateau, where satellite monitoring can make up such deficiency. In this study, by a combination of AMSR-E (2002–2011) and AMSR-2 (2012–2021) passive microwave data, MODIS optimal products and in situ measurements of temperature profiles in four lakes, the ice phenology events of 40 high-elevation large lakes were derived and their inter-annual trends and influencing factors were analyzed. The freeze-up start date (FUS) mainly occurs in November-December with an average date of 9 December and the break-up end date (BUE) is concentrated in April-May with a multi-year average of 5 May. Under climate warming, 24 of the 34 (70.6%) lakes show delayed FUS at an average trend of 0.35 days/year, and 7 (20.6%) lakes show advanced BUE (rate of change CR = −0.17 days/year). The average ice coverage duration (ID) was 147 days, and 13 (38.2%) lakes shortened ID at an average rate of −0.33 days/year. By synthesizing other ice phenology products, we obtained the assembled products of lake ice phenology, and found that air temperature dominates during the freeze-thaw process, with a higher dependence of BUE than that of FUS on air temperature.

## 1. Introduction

Lake ice, as an important component of the cryosphere, has important implications for local and global climate, energy balances, and hydrological cycles. For example, longer ice-off conditions increase the heat transfer, the annual evaporation, and the greenhouse gases emissions from lakes due to chemical and biological processes [1]. In addition, the cooling effect of latent heat flux through evaporation can affects regional climate, such as lake-effect clouds and precipitation [2,3]. At the same time, lake ice is sensitive to changes in climate [4]. Some studies have suggested that lake ice phenology is a good proxy for climate change, and both freeze-up and break-up dates can represent air temperature change [5,6,7]. Moreover, in some case, lake ice records may be a more reliable climate indicators than air temperature records [8,9]. However, it is very difficult to conduct high-density and continuous artificial lake ice observations. The development of remote sensing technology has made it possible to monitor lake ice continuously on a regional and global scale [1,10,11] and the observation of lakes from space will expand our global understanding of lake responses to a changing climate.

Optical remote sensing has been widely used for ice phenology monitoring because of its high temporal and spatial resolutions and that optical images are easily accessible; however, it is a passive system, which can be limited by cloud cover and daylight hours [12]. The presence of clouds over the lake is not conducive to determining the date of freezing and thawing [13] and is easily confused with ice in current algorithms [14]. The occurrence of polar night in high latitude will limit the application of optical remote sensing [15]. The common method for determining lake ice phenological parameters by optical remote sensing are the visual interpretation of remote sensing images [16], automatic thresholding method based on reflectance spectral curve [1,17], and the threshold-based classification method combined with lake ice/water proportions [18,19]. Compared with optical remote sensing, microwave remote sensing has advantages of penetrating clouds, is less affected by weather conditions and lighting, and is more conducive to monitoring the dynamic changes of lake ice. Among them, active microwave remote sensing has high spatial resolution, and can select different wavelength and emission modes according to different ground object attributes to obtain the required information. However, radar signals can be affected by wind, bubbles and/or locally drifted snow, leading to the misidentification of ice from water during freeze-thaw process [10]. Furthermore, limited global coverage and temporal frequency of observation has constrained the application of active microwave remote sensing in global lake ice monitoring [11,12]. Lake ice monitoring with active microwave mainly adopts a threshold method [10,20], an unsupervised classification segmentation method [21], or a machine learning method [22,23]. Spaceborne microwave radiometers have observations with relatively high temporal resolution since 1978, which are valuable to ice phenology studies [11]; however, its coarser spatial resolution is only suitable for the study of great lakes. Passive microwave extracts lake ice phenological parameters using its brightness temperature with one method in extraction of sudden changes in brightness temperature curve [11,24,25,26,27], and the other in obtaining polarization ratio (PR) and frequency gradient (GR) to determine the freezing and thawing of the entire lake surface [28,29]; however, obvious differences in derived results of lake ice phenology events by different satellite data or by different retrieving methods exist. For example, the difference in the average freeze-up start date in Dogai Coring can reach 39 days [30,31], and that of Xijir Ulan Lake can reach 58 days [18,32]. Currently, no studies have systematically analyzed these differences. In this paper, eight lake ice phenology datasets were integrated to reveal the causes of these differences and to obtain the average lake ice phenology events for 40 high-elevation large lakes on the Tibetan Plateau.

The application of different remote sensing data and methods has filled and expanded lake ice monitoring records, and the results of lake ice phenology trend analysis reveal a general phenomenon that under the background of global warming, most of the lakes will freeze up later, break up earlier, and have shorter ice coverage duration. For example, Du et al. [11] used AMSR-E/2 brightness temperature data during 2002 to 2015 to analyze the ice phenology changes of 71 large lakes in the northern hemisphere and found that 49.3% of the lakes had later freeze-up end dates, 56.3% of the lakes had earlier break-up end dates, and 60.6% of the lakes had shorter lake ice coverage duration. Because of the complexity of the influencing factors in the freeze-thaw process, some lakes have abnormal characteristics of lake ice phenology (earlier freeze-up start date, later break-up end date, longer ice coverage duration), such as Lake Ladoga, and Lake Baikal [28,33,34]. The influencing factors of lake ice phenology include climatic factors (temperature, precipitation, wind, radiation, etc.) and non-climatic factors (lake depth, altitude, inflow, etc.), and each factor can change the formation and ablation of lake ice by affecting the heat transfer process [35]. The dominant influencing factors were different in lakes [33,36,37], thus resulting in the complexity of lake ice phenology variations.

The Tibetan Plateau (TP) has the greatest total lake area in China making up 1.4% of its total area [38]. The dynamic change of lake ice will affect the heat exchange between the surface and the atmosphere, and then cause the change of the regional or even the whole TP heating [39,40]. Studies have shown that the thermal effect of the TP is an important reason for its influence on the atmospheric circulation and the climate of Asia and even the world [41,42,43,44]. At the same time, the TP was identified as one of the most sensitive regions in the world to changes in climate [45]. However, due to the lack of ground observation, studies on the impact of global climate change on the TP is still limited. Lake ice phenology retrieved from remote sensing data, as an effective indicator of climate change, can fill a gap in our knowledge about the impact of climate change in this remote region. Many lake ice phenology datasets on the TP have been developed [18,30,31,32,46,47,48,49,50,51,52]; however, the uncertainties in multiple remote sensing data, multiple methods, and multiple definitions of lake ice phenology parameters in previous studies, make the consistency between lake ice phenology datasets very weak. In this study, based on the passive microwave remote sensing data and in situ measurements, we summarize and analyze the lake ice phenology of the Tibetan Plateau. The contents are as follows: Section 2 introduces the study area and methods, including the introduction of data, and the extraction method of ice phenology events, etc.; Section 3 introduces the integrated analysis of eight lake ice phenology datasets, the influencing factor analysis, the cluster analysis, and the trends of lake ice phenology events during 2002–2021; Section 4 discusses the influences of snow cover on the lake ice phenology events and their trends, the effect of pixel position, and the source of errors existing in this study; Section 5 documents the conclusions.

## 2. Study Area and Methods

Tibetan Plateau (26°00′12″ N–39°46′50″ N, 73°18′52″ E–104°46′59″ E), known as “the Third Pole of the earth”, is located in the southwest of China, and has an area of 2572.4 × 10^3^ km^2^ [53]. The Tibetan Plateau is rich in water resources, and lakes with an area of 1 km^2^ and above account for 57.2% of the total lake area in China [54]. In order to adapt to the spatial resolution of passive microwave remote sensing data, this study selected 40 medium and large lakes for ice phenology research (Figure 1, Table 1). The ASTER Global Digital Elevation Model V003, which has a horizontal resolution of 30 m and can be obtained from NASA EARTHDATA (https://www.earthdata.nasa.gov/, accessed on 8 January 2022), was used to obtain the attribute information (area, altitude, latitude, and longitude) of the targeted lakes (Table 1).

### 2.1. Data

#### 2.1.1. Remote Sensing Data (MODIS, AMSR-E, AMSR-2)

The TERRA and AQUA satellites were successfully launched in 1999 and 2000, respectively, with complementary data collection times and being called as “Earth Observation First Morning Star” and “Earth Observation First Afternoon Star”. The satellite is equipped with a Moderate-resolution Imaging Spectroradiometer (MODIS), which has 36 bands, of which 2 bands have a resolution of 250 m, 5 bands are 500 m, and the remaining 29 bands are 1000 m [55]. This study selected MODIS 1B products (MOD02HKM and MYD02HKM) (https://www.earthdata.nasa.gov/, accessed on 25 March 2022) with a resolution of 500 m after radiometric calibration, and the lake ice was visually discriminated according to a false-color composite image formed by the red band (B1), the near-infrared band (B2), and the blue band (B3).

The passive microwave imager AMSR-E (The Advanced Microwave Scanning Radiometer for EOS) and AMSR-2 (The Advanced Microwave Scanning Radiometer 2) were launched with the Aqua satellite and the Global Change Observation Mission 1st-Water satellite respectively, which provide vertical and horizontally polarized brightness temperature observations with spatial resolutions ranging from 3 × 5 km (89.0 GHz) to 75 × 43 km (6.925 GHz). The G-Portal (https://gportal.jaxa.jp/gpr/, accessed on 4 January 2022) provides Level-3 brightness temperature products of AMSR-E and AMSR-2. In this study, the level-3 horizontal polarization brightness temperature data of 18 GHz was selected. The projection method was equidistant columnar Projection (EQR) with a grid accuracy of 0.1°. The data periods selected in this paper were: 1 August 2002–31 July 2011 (AMSR-E), 1 August 2012–31 July 2021 (AMSR-2), among which there are no data for some periods (1–7 August 2002, 13–19 September 2002, 30 October–5 November 2003, 19 November 2004, 18 November 2006, 28 November 2007, 3–4 February 2010, 1–2 July 2012, 11–13 May 2013).

#### 2.1.2. Long-Term Series of Daily Snow Depth Dataset in China (1979–2021)

Daily snow thickness distribution data in China stretch from 1 January 1979 to 31 December 2020, with a spatial resolution of 25 km [56,57,58,59]. This dataset was used to calculate the average snow days with a snow depth of ≥5 cm from December to May in five regions of the TP from 2002 to 2020 (Figure 1), focusing on the impacts of snow cover on the break-up end date.

#### 2.1.3. ERA5-Land

ERA5-Land is a reanalysis dataset provided by the European Centre for Medium-Range Weather Forecasts. It provides data on water and energy cycles from 2.89 m below ground to 2 m above the surface since 1950, with a horizontal resolution of 0.1° (9 km) and a temporal resolution up to 1 h. This study uses the ERA5-Land monthly average data provided by the Copernicus Climate Change Service Data Platform from 1950 to the present (https://cds.climate.copernicus.eu/, accessed on 1 April 2022), and selects the monthly average data of 2 m air temperature, 10 m wind speed, surface downward shortwave radiation and snow depth from 2000 to 2021 to analyze the influencing factors of ice phenology events. 

#### 2.1.4. Lake Temperature Observation

In Nam Co [60,61,62], Peiku Co [63], Bangong Co, and Daze Co [64], the vertical temperature gradient measurements have been collected. The observation period of Nam Co is from November 2011 to June 2014 (daily), the observation period of Peiku Co is from June 2016 to May 2017 (hourly), and the observation period of Bangong Co is July 2012 to August 2013 (hourly), the observation period of Daze Co is from August 2012 to August 2013 (hourly). The lake temperature observations can provide in situ datasets for judging the freezing and thawing dates of lakes (as shown in Appendix A): take the first minimum point of water temperature lower than the maximum density temperature as the freeze-up start date, the minimum water temperature point as the freeze-up end date, the point where the water temperature starts to rise rapidly and the stratification is obvious as the break-up start date, and the point where the water temperature first peaked after the break-up start date is the break-up end date.

### 2.2. Methods

#### 2.2.1. Extraction of Lake Boundaries

After the DEM was projected, the continuous area with slope 0 was identified as a lake to determine the lake boundary. The lake boundaries of HydroLAKES [65] were used to verify the extraction results, and lakes with an area >200 km^2^ were selected as the target lakes (Figure 1 and Table 1).

#### 2.2.2. Extraction of Lake Ice Phenology Events

Phenology is the field of science concerned with cyclic and seasonal natural phenomena, especially in relation to plant and animal life and climate. In glaciology, freshwater ice scientists refer to freeze-up/break-up and duration of ice on lakes and rivers as ice phenology [66]. Lake ice freeze-up start (FUS) corresponds to the date when detectable ice appears. Freeze-up end (FUE) is defined as the date when the lake is fully ice-covered. Break-up start (BUS) is the first day of the year on which a detectable ice-free water surface appears. Break-up end (BUE) is defined as the date on which a lake is completely clear of ice. Time between FUS and FUE determines the freeze-up duration (FUD), while time between BUS and BUE determines the break-up duration (BUD). Lake ice cover duration (ID) is defined as the period between FUS and BUE, and complete ice cover duration (CID) is defined as the period between FUE and BUS.

In this paper, the lake ice phenological parameters were determined by the brightness temperature curve of the central lake pixel. The linear interpolation method was used to complete the missing brightness temperature data, and the median filtering method was used to denoise the brightness temperature sequence. The brightness temperature difference search method [24] combined with temperature threshold was used to automatically extract FUE and BUS dates. The specific discrimination algorithm is shown in Equations (1) and (2).
(1)$$\mathrm{FUE}=\min(\left({\mathrm{TB}}_{i}+{\mathrm{TB}}_{i-1}+{\mathrm{TB}}_{i-2}+{\mathrm{TB}}_{i-3}\right)/4-\left({\mathrm{TB}}_{i}+{\mathrm{TB}}_{i+1}+{\mathrm{TB}}_{i+2}+{\mathrm{TB}}_{i+3}\right)/4),$$
(2)$$\mathrm{BUS}=\max(\left({\mathrm{TB}}_{i}+{\mathrm{TB}}_{i-1}+{\mathrm{TB}}_{i-2}+{\mathrm{TB}}_{i-3}\right)/4-\left({\mathrm{TB}}_{i}+{\mathrm{TB}}_{i+1}+{\mathrm{TB}}_{i+2}+{\mathrm{TB}}_{i+3}\right)/4),$$
where, ${\mathrm{TB}}_{i}$ is the filtered brightness temperature on the ith day, i = 1, 2,…, 365 or 366, and min () and max () are the minimum and maximum functions. The maximum and minimum values of the time series of the brightness temperature difference were taken as the abrupt change points of the brightness temperature curve.

The positive and negative curve intersection method [25] combined with the brightness temperature threshold was used to determine FUS and BUE. The positive and negative curves are shown in Equations (3)–(6). Curves Y1 and Y4 were used to determine FUS, and curves Y2 and Y3 were used to determine BUE; the method is described in more detail in the Appendix A.
(3)$$Y1=\left({\mathrm{TB}}_{i}+{\mathrm{TB}}_{i-1}+{\mathrm{TB}}_{i-2}+{\mathrm{TB}}_{i-3}\right)/4-\left({\mathrm{TB}}_{i}+{\mathrm{TB}}_{i+1}+{\mathrm{TB}}_{i+2}+{\mathrm{TB}}_{i+3}\right)/4+1,$$
(4)$$Y2=\left({\mathrm{TB}}_{i}+{\mathrm{TB}}_{i-1}+{\mathrm{TB}}_{i-2}+{\mathrm{TB}}_{i-3}\right)/4-\left({\mathrm{TB}}_{i}+{\mathrm{TB}}_{i+1}+{\mathrm{TB}}_{i+2}+{\mathrm{TB}}_{i+3}\right)/4-1,$$
(5)$$Y3=\left({\mathrm{TB}}_{i}+{\mathrm{TB}}_{i+1}+{\mathrm{TB}}_{i+2}+{\mathrm{TB}}_{i+3}\right)/4-\left({\mathrm{TB}}_{i}+{\mathrm{TB}}_{i-1}+{\mathrm{TB}}_{i-2}+{\mathrm{TB}}_{i-3}\right)/4+1,$$
(6)$$Y4=\left({\mathrm{TB}}_{i}+{\mathrm{TB}}_{i+1}+{\mathrm{TB}}_{i+2}+{\mathrm{TB}}_{i+3}\right)/4-\left({\mathrm{TB}}_{i}+{\mathrm{TB}}_{i-1}+{\mathrm{TB}}_{i-2}+{\mathrm{TB}}_{i-3}\right)/4-1,$$

After lake ice phenology parameters were automatically extracted, the visual interpretation of the brightness temperature curve was used to adjust the unreliable results.

#### 2.2.3. Statistical Analysis

The Mann–Kendall trend analysis [67,68] method was used to analyze the extracted FUS, BUE, ID during 2002–2021. As there is no passive microwave brightness temperature data from 1 August 2011 to 31 July 2012, the freeze-thaw parameters in this year were replaced by the multi-year average values. In the Mann–Kendall trend test, the positive (negative) statistic Z is used to represent the increasing(decreasing) trend, and the size of |Z| represents the significance of the trend change. If |Z| ≥ 1.28, the trend characteristics are significant, and the confidence level is ≥ 90%. The trend values are characterized by the slope β. In this paper, the statistic Z was used to analyze the trend change, and the slope β was used to analyze the rate of change.

In the integrated analysis of the eight lake ice phenology datasets, the deviation of each dataset was measured by the root mean square error (RMSE, Equation (7)), where the subscript i is the ith lake (i = 1,2,…,40), n is the total number of lakes (n = 40), j is the jth lake ice phenology dataset (j = 1,2,…,8), $y_{\mathrm{ij}}$ is the ice phenology dates of the ith lake in the jth dataset, and ${\bar{y}}_{i}$ is the average ice phenology of the ith lake calculated from multiple datasets. ${\mathrm{RMSE}}_{j}$ is RMSE of the jth dataset.
(7)$${\mathrm{RMSE}}_{j}=\sqrt{\frac{1}{n}\sum_{i=1}^{40}{\left(y_{\mathrm{ij}}-\bar{y_{i}}\right)}^{2},}$$

In order to find the dependence of the lake average freezing and thawing dates (Table 1) on climatic factors (temperature, wind speed, snow depth, radiation) and local factors (latitude and longitude, area), correlation analysis was carried out, and a *t*-test was used to test the significance of the correlation coefficient (r). Finally, cluster analysis on the lakes was conducted based on the most important factors.

## 3. Results

### 3.1. Verification and Assessment of Lake Ice Phenology Datasets

The extracted lake ice phenology events indicate that some lakes were not frozen/completely frozen during 2002–2021 (as shown in Appendix A). To illustrate the reliability of the results, MODIS false-color composite images during ice covered seasons were selected to verify the unfrozen conditions. The MODIS false-color image show consistency with the results obtained from the AMSR-E/2 data (as shown in Appendix A), wtih Dogai Coring not completely frozen in 2002–2003 (Appendix A), Paiku Co not frozen in 2016–2017 (Appendix A), Xuru Co not frozen in 2005–2006 (Appendix A), Taro Co not completely frozen in 2002–2003 (Appendix A), and Tangra Yumco not completely frozen in 2015–2016 (Appendix A). Xijir Ulan Lake is the only extra, where the lake was defined as not completely frozen by the MODIS composite image, but as not frozen by AMSR-E/2. This may be probably related with the position of the selected microwave pixel, check Section 4.2 for more content.

The estimated ice phenology events from the satellite datasets and the lake temperature profiles (as shown in Appendix A) are summarized in Table 2; the FUE and BUE dates obtained by satellite products are close to the results obtained by the lake temperature profiles, with high consistency regardless of the size of the lake. In contrast, the FUS and BUS dates obtained by the lake temperature profile are earlier than the satellite products, where ice forming or melt can happen at shallower depths, but satellite products can only sense the status of the lake surface at deeper depths because of its coarse resolutions. For example, from 2012 to 2013 in Dagze Co, the biases of FUE and BUE dates in most of studies were within 1 and 2 days respectively, while that of FUS and BUS dates ranged from 16 to 25 days and 37 to 47 days, respectively. This similar phenomenon with relatively small (large) biases in FUE and BUE (FUS and BUS) dates also occurred in Bangong Co and Nam Co. 

The observation data of Paiku Co from 2016 to 2017 show that the lake was in a relatively uniform state of mixing throughout the winter and no frozen ice appeared (as shown in Appendix A). The MODIS image on 19 Mar 2017 proves this situation (as shown in Appendix A); however, the dates of freezing and thawing were determined by Guo et al. [51], who used model simulations. 

Further, the multi-source average of each lake ice phenology parameter was used as the standard, and the root mean square error (RMSE) was used to evaluate the deviation of each dataset. Overall, the results from Qiu [47] and Cai et al. [50] are closer to the average lake ice phenology (RMSE are both within 6 days). Although this study was not the best when compared with the average results, our estimation shows some advantages when compared with that of lake temperature profiles (Table 2). For example, the estimated FUS, FUE (BUE) dates in Nam Co during 2012–2013 (2013–2014) in this study were the closest to the estimation from the lake temperature profiles. The errors of FUE and BUE dates between our estimation and those estimated through lake temperature profiles were all within 5 days. These indicate that the lake ice phenology parameters determined in this study are reliable.

### 3.2. Comprehensive Integrated Analysis of Lake Ice Phenology Datasets

The derived lake ice phenology events by our method were integrated with the results of seven other studies [18,30,31,32,47,50,51] (Figure 2a–d), and the average freeze-thaw dates of the 40 lakes in the TP region were obtained (Table 1).

The boxplot shows the magnitude of the difference between the datasets. Through the different remote sensing data types, the FUS determined by passive microwave is generally later than the results obtained by MODIS (Figure 2a), and this can be attributed to the freezing and thawing patterns of lakes. Many lakes begin to freeze in the shallow water near the lakeshore. The MODIS pixels with higher spatial resolution cover the entire lake surface, and this change is more easily observed, while passive microwave pixels, with a coarse resolution and locating at the center over deep water, have less interference from the surrounding land surface. Differences in the spatial distribution of pixels lead to differences in the observed timing of lake ice formation, which is similar to the results of Cai et al. [19,50]. In addition, the initially formed thin ice may not be easily detected by the brightness temperature threshold [11], resulting in a later FUS determined by passive microwave data. Meanwhile, most of the FUE (BUS) dates obtained by passive microwave are earlier (later) than MODIS (Figure 2b,c), which may be since the lake ice ratio corresponding to the brightness temperature threshold is smaller than that used by MODIS to determine lake ice phenology parameters. Moreover, compared with the FUS, FUE, and BUS dates, the consistency of the BUE date is better, and there is no particularly large difference (Figure 2d), which is consistent with the conjecture of Cai et al. that the impact of different remote sensing data on the freezing date may be greater than that on melting dates [50]. However, in general, the BUE date extracted by passive microwave is earlier than that of MODIS data, which may be related to the earlier ice melting in deep water. The calculation of duration also reflects the differences in remote sensing data: the FUD date, BUD date, CID, and ID obtained by passive microwave data are mostly shorter than those obtained by MODIS data (as shown in Appendix A).

The above analysis shows that there are differences in lake ice monitoring by using different remote sensing data, and the multi-sources average can balance the differences and obtain general lake ice phenological results (Table 1). The FUS of lakes on the Tibetan Plateau is mainly concentrated in November and December. Among them, Hoh Xil Lake and Ulan Ul Lake began to freeze at the earliest (27 October), while Xuru Co began to freeze at the latest (29 January), and arrived in late January of the following year, all lakes were in a completely frozen state (Nam Co was completely frozen at the latest, 29 January). In late February of the following year, the lakes began to melt one after another. Among them, Taro Co began to melt at the earliest time (19 February). By April and May, the lake ice of most lakes disappeared completely, but the ice in a very few lakes could last until June and July. The ice coverage duration varies widely, from 79 days (Xuru Co) to 239 days (Hoh Xil Lake).

### 3.3. Analysis of Influencing Factors of Lake Ice Phenology Parameters

We carried out a statistical analysis of the dependence of the average ice phenology parameters (Table 1) on climatic factors and local factors (Table 3) in 40 lakes on the TP. The analysis results showed that temperature had the most significant effect on the freeze-thaw dates. The correlation coefficient between “average temperature in December” and FUS can reach 0.77, and the correlation between “average temperature in June” and “BUE” was higher (r = −0.79). This confirms that lake phenology was highly determined by thermodynamic factors as reported by Kouraev et al. [33], with BUE more thermally determined [66]. There were high correlations between freeze-thaw dates and latitude (r = −0.71 for FUS and r = 0.41 for BUE), solar radiation (r = 0.54 for FUS and r = −0.16 for BUE), and altitude (r = −0.1 for FUS and r = 0.38 for BUE), which is attributed to the autocorrelation between these factors and temperature. In general, low altitudes, low latitudes and strong solar radiation correspond to high air temperatures. The correlation between snow cover and freeze-thaw dates were significant (r = −0.41 for FUS and r = 0.29 for BUE), because snow accumulation can affect the growth and thaw rates of the ice cover by adding insulation and albedo. Snowfall is beneficial to the initial formation of lake ice and slow down the melting process of lake ice, which is consistent with previous studies [35,69,70,71,72,73]. The correlation between wind speed (in December) and freeze-thaw dates was small (r = −0.21 for FUS), mainly because the freezing and thawing of lakes is a fast-changing process, which can be completely frozen or thawed in about half a month, while the monthly wind speed contains too much information and does not reflect the characteristics of wind speed changes in the days before and after FUS and BUE. Overall, air temperature dominated during the freeze-thaw process, and BUE had a stronger dependence on air temperature than FUS, and snow was important in the freeze-thaw process.

Through the above correlation analysis, the most important factors for lake ice phenology (elevation, latitude, area, average temperature in December, average temperature in June, shortwave radiation, and snow depth) were selected to perform cluster analysis on the lakes. The ice phenology dates of each group of lakes are a reflection of climate and lake attribute characteristics (as shown in Appendix A).

### 3.4. Trend Analysis of the Lake Ice Phenology Extracted by AMSR-E/2 from 2002 to 2021

Among the 40 lakes, six (Dogai Coring, Xijir Ulan Lake, Taro Co, Paiku Co, Tangra Yumco, Xuru Co) have unfrozen/incomplete freezing conditions, thus they are not considered in trend analysis. Among the remaining 34 lakes, 24 (70.6%) had later FUS and 12 were statistically significant at the 90% confidence level. The average rate of change in later FUS was 0.35 days/year, while only 7 (20.6%) lakes had earlier BUE at an average trend of −0.17 days/year. 

Ice cover duration (ID) was affected by both FUS and BUE. Among them, 16 (47.1%) lakes extended the ID at an average rate of 1.17 days/year, 13 (38.2%) lakes shortened ID at an average rate of −0.33 days/year, and 5 (14.7%) lakes had no obvious trend in ID (Table 4). The above results do not seem to be consistent with the changing characteristics of the earlier BUE and the shorter ID under the background of global warming, and further discussion can be found in Section 4.1.

## 4. Discussion

### 4.1. Effects of Snow Cover on Lake Ice Phenology and Trend Change

In order to understand the reasons for the later BUE of most lakes, the BUE variation curves of lakes from 2002 to 2021 were drawn. Taking Qinghai Lake as an example (Figure 3), the BUE of the last few years (2018–2021) was later than other years, which obviously increases the positive trend of BUE, and this phenomenon generally exists in Har Lake, Kusai Lake, Zhuonai Lake, Serling Co and other lakes. To illustrate the influence of the BUE anomaly from 2018 to 2021 on the overall change trend, the change trend of BUE of each lake from 2002 to 2018 was calculated (Table 5). Compared with 2002–2021 (Table 4), more lakes had later BUE and shorter ID from 2002 to 2018 (Table 5), and the proportion of earlier BUE increased from 20.6% to 41.2%. The later FUS is still common (76.5%), and correspondingly, more lakes had shorter ID with proportion increased from 38.2% to 64.7% (Table 6). The trend analysis results (2002–2018) are consistent with Cai et al. They used MODIS snow products to study the phenological changes of lake ice in 58 lakes in the TP region from 2000 to 2017, and found that 81.03% of the lakes have later FUS, 50% of the lakes have earlier BUE, and 68.79% of lakes have shorter ID [50]. Liu et al. [74] investigated the changing characteristics of lake ice phenology over the TP during 2002–2015 and found that the lakes in the southern TP had delayed break-up dates and prolonged ice durations, which was attributed to the impact of the winter North Atlantic Oscillation (NAO). Both studies indicate that the trend toward earlier BUE need not be dominant.

The delay of BUE is likely to be related to the increase of snow cover [69,71]. To confirm this idea, a long-term series of daily snow depth data in China (1979–2021) was selected for verification. From 2002 to 2020, the average snow days with a snow depth ≥ 5 cm from December to May of the next year in each region were drawn (Figure 1 and Figure 4). It can be clearly seen that 2018 and 2019 were snowy years, which correspond to the later BUE in 2018 and 2019. Therefore, the trend towards later BUE in most lakes from 2002 to 2021 is likely to be caused by snow cover: from 2018 to 2020, the abnormal increase in snow cover led to abnormally late BUE, which increased the trend of delaying BUE, thus covering up the trend of advancing BUE.

### 4.2. Effect of Pixel Position on Lake Ice Phenology Parameters

It is worth noting that the difference in pixel position also affects the results of lake ice phenology. Taking Nam Co as an example, two pixels were selected in the east and west of the lake (the distance between the two is about 30 km), and the lake ice phenology parameters from 2002 to 2021 (except 2011 to 2012) were extracted respectively (as shown in Appendix A). The pixel position will affect the judgment of whether the lake is completely frozen. The difference of break-up dates caused by the difference of pixel position was also obvious, which can reach up to 20 days, and the break-up dates of eastern pixel were later than that of the western pixel (Figure 5).

### 4.3. Uncertainty Analysis

Insufficient spatial and temporal resolution of remote sensing data and the limitations of the methods brought uncertainties in the extraction of lake ice phenology parameters. In this study, the absence of remote sensing data will directly affect the extraction results. In addition, the coarse spatial resolution (0.1°) will result in the brightness temperature curve being contaminated by lakeshore radiation, while the degree of influence of radiation pollution on the results cannot be quantified. The limitations of the brightness temperature difference search method have been pointed out in some other studies [26,27]; when the jumping phenomenon of high platform values on the brightness temperature curve is not obvious, the results obtained through the method of searching for the maximum and minimum are most likely not located at the initial position of high platform values, which may also appear in our estimation. As for some unfrozen/not completely frozen lakes, they cannot be automatically identified by the program, and can only be judged by the method of visual interpretation of the brightness temperature curve.

The available time series of AMSR-E/2 data allowed us to derive trends in lake phenology parameters for the studied lakes, however, the results of trend analysis are easily affected by fluctuations in individual years due to short time series. For example, advanced BUE events appear more when heavy snow years are removed. Therefore, a combination of satellite products (optical and passive microwave data) and numerical simulations is suggested for long-term trend analysis.

In the process of merging the eight lake ice phenology datasets, the results of lake ice phenology derived from different remote sensing data were very different. For example, the average FUS date of Dogai Coring determined by optical remote sensing and passive microwave remote sensing can differ by 36 days [30,47]. The average FUE (BUS) date of Migriggyangzham Co (Xijir Ulan Lake) can differ by 30 (74) days [18,31]. As five of the eight datasets came from MODIS products, the average results will contain more optical information and the RMSEs of passive microwave data are larger. In addition, the study period, and the number of lakes in each dataset were different. Consequently, lakes with more datasets got more representative averages, while lakes with fewer datasets were more susceptible to individual results.

The frozen-thaw process will be affected by atmospheric heat exchange, water heat storage, solar radiation, ice and snow conditions, wind, inflow, etc. [69]. However, water depth as the function of lake heat storage was not considered in this study. Although the effects of wind and radiation were considered, the data used do not reflect changes in wind speed around freeze-thaw dates and the inter-annual variations in radiation.

Although there were many uncertainty factors in this study, the results are relatively reliable. The trend analysis results (2002–2018) are consistent with Cai et al. [50], but there were large differences in the results of individual lake ice phenology parameters, For example, the BUE of Lexiewudan Co during 2002–2003 extracted by Cai et al. was 11 July 2003, while the BUE obtained in this study was 27 April 2003, with a 75-day difference. Nevertheless, the MODIS false color composite images (as shown in Appendix A) showed that Lexiewudan Co had completely melted ice on 28 April 2003, indicating that Cai et al. still had certain errors in the extraction of BUE for Lexiewudan Co. The choice of study area has great influence on the results of trend analysis. Yao et al. [32] studied the changes of lake ice in 20 lakes in the Hoh Xil region from 2000 to 2011, and their results were quite different from this study. They found that 90.9% of lakes had later FUS (change rate = 0.73 days/year), 86.4% of lakes had earlier BUE (change rate = −0.81 days/year), and 90.9% of lakes had shorter ID (change rate = −1.91 days/year). Compared with this study, the lakes in the Hoh Xil region had higher proportion with earlier BUE and faster change rate. This indicates that the lakes in the Hoh Xil region respond strongly to climate change. However, although there are similar research areas, the change rate of the ID obtained in this study was less than other research results [18,30]. Wang et al. studied 40 lakes in the TP region from 2000 to 2015, and found that the average change rate of ID was 0.47 days/year [30], while in this study the average change rate of ID from 2002 to 2018 is −0.05 days/year. Kropáček et al. [18] studied 59 lakes in the TP area from 2001 to 2010. They clustered the lakes by considering the lake climatic conditions and local conditions, and the C group lakes (as shown in Appendix A) in this study were similar to their results. In this study, the average change rate of ID from 2002 to 2018 was −0.53 days/year, while that can reach −1.6 days/year in Kropáček et al.’s study. Differences in durations, extraction methods and the lakes may be the reasons.

## 5. Conclusions

This paper uses the AMSR-E/2 passive microwave brightness temperature data from 2002 to 2021 (except 2011 to 2012) to extract lake ice phenology parameters for lakes in the TP region, and the MODIS 1, 2 and 3 band false color composite images were used to verify the unfrozen/incompletely frozen lakes, and the two had good consistency. An analysis of the lake ice phenology trend of 34 lakes that were completely frozen in each of year from 2002 to 2021 found that 70.6% of the lakes had a later FUS, and only 20.6% of the lakes had earlier BUE. Considering that snow cover may be the reason for the abnormal trend of BUE, the last few snowy years (2018–2020) were eliminated and the trend analysis was carried out, and it was found that the average FUS of 34 lakes in 2002–2018 was 4 December, the trend towards later FUS was still common (76.5% of the lakes, average change rate = 0.59 days/year), and a few lakes (23.5%) have a trend towards earlier FUS (average change rate = −0.68 days/year). The average BUE was 7 May, and the proportion of lakes with earlier BUE increased to 41.2% (average change rate = −0.38 days/year), but lakes with a lagging trend in BUE still dominated (52.9%, average change rate = 0.79 days/year). The average ID affected by FUS and BUE was 153 days, 64.7% of lakes had shorter ID (average change rate = −0.76 days/year), and 35.3% of lakes had longer ID (average change rate = 1.24 days/year) (Table 6).

Comparing the results of this study with the other seven lake ice phenology datasets, it was found that the type of remote sensing data has an influence on the results of lake ice phenology parameters. In general, the FUS, BUS (FUE, BUE) dates extracted from passive microwave data were later (earlier) than the extraction results of MODIS data. The FUD date, BUD date, and CID and ID values of lakes determined by passive microwave data were almost shorter than the results from the MODIS data, but the specific differences will also be affected by the algorithm, lake properties, etc. 

The lake temperature profile measurements provide a new validation dataset for the determination of lake ice phenological parameters. The FUE and BUE dates determined by lake temperature were in good agreement with remote sensing results, but FUS and BUS dates were significantly earlier than those derived from remote sensing. In order to balance the influence of remote sensing data types and algorithms on the extraction results of lake ice phenology parameters, the results of each dataset were synthesized to obtain the average ice phenology results of lakes in the TP area. It was found that the TP lakes mainly began to freeze in November and December, and the lake ice completely melted in April and May of the following year. However, the freezing and thawing dates of lakes vary greatly, mainly due to different meteorological conditions and local factors. Through the correlation analysis of lake ice phenology influencing factors, it is found that temperature is dominant, and the dependence of BUE date on temperature is higher than that of the FUS date. Considering several factors (elevation, latitude, area, average temperature in December, average temperature in June, shortwave radiation, and snow depth) that have a greater impact on lake ice phenology, cluster analysis of lakes found that the lakes in high altitude, high latitude and low temperature areas have the characteristics of early freezing, late melting, and long lake ice duration.

Although the application of deep learning and machine learning algorithms in lake ice research is very limited, its advantages have been shown. For example, the accuracy performance of convolutional neural network (CNN) is greater than 92% [75]. These automatic classification methods are more suitable for lake ice identification than previous methods (such as the threshold method). Therefore, advanced data processing methods should be introduced in the future study of lake ice phenology on the TP, such as, k-means algorithm, Iterative Region Growing and Semantics (IRGS) algorithm, support vector machine (SVM) algorithm, the genetic algorithm (GA), and convolutional neural network (CNN), etc.

## Figures and Tables

**Figure 1 sensors-23-01661-f001:**
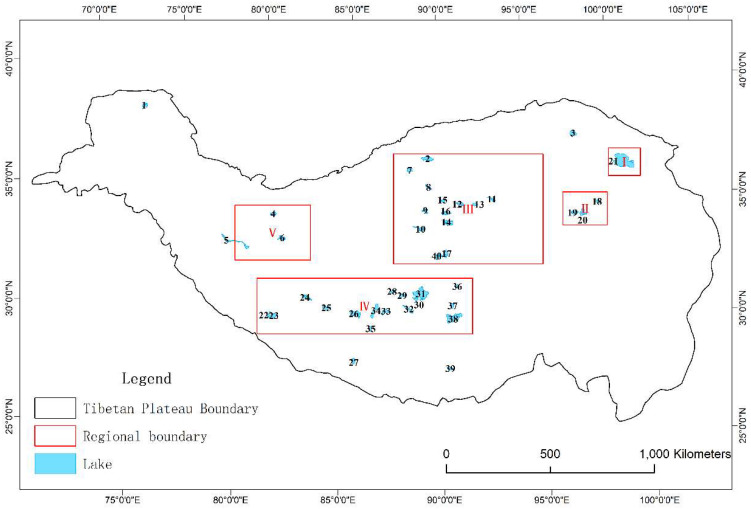
Distribution of the targeted lakes on Tibetan Plateau, where the black numbers indicate the positions of the lakes, with lake information shown in Table 1.

**Figure 2 sensors-23-01661-f002:**
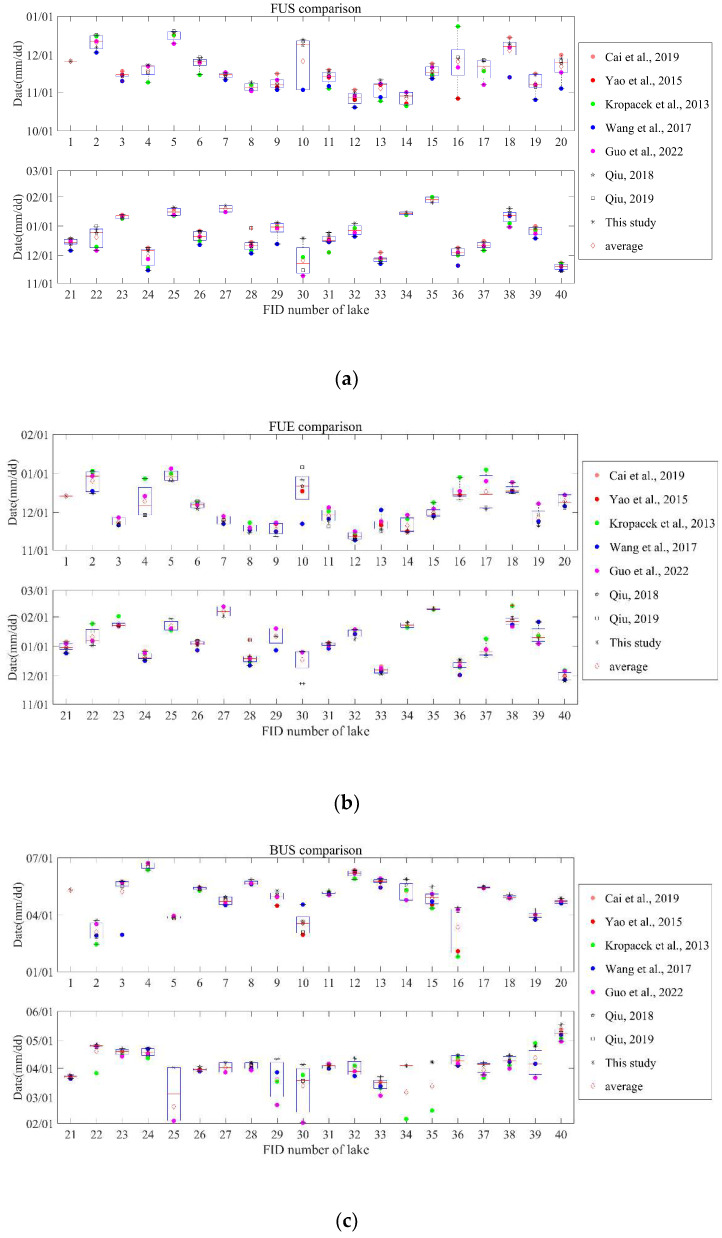
(**a**) Average freeze-up start date (FUS) of 40 lakes in different datasets. (**b**) Average freeze-up end date (FUE) of 40 lakes in different datasets. (**c**) Average break-up start date (BUS) of 40 lakes in different datasets. (**d**) Average break-up end date (BUE) of 40 lakes in different datasets. The solid-colored dots are the results obtained from MODIS data, the hollow figures with black borders are the results obtained from passive microwave data, and the red diamonds are the multi-source average of all the results. FID numbers are shown in Table 1 [18,30,31,32,47,50,51].

**Figure 3 sensors-23-01661-f003:**
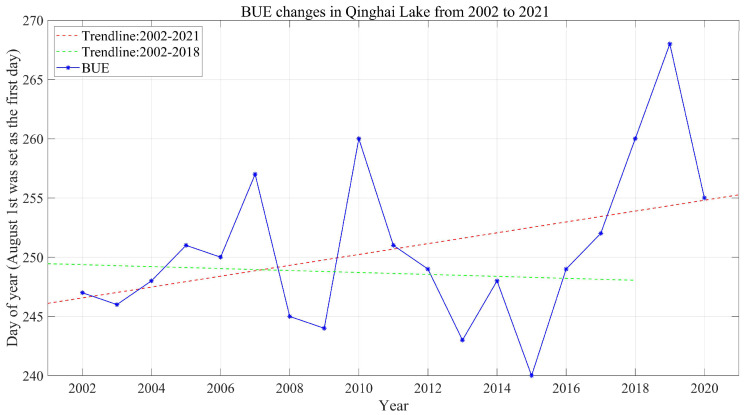
Break-up end date changes in Qinghai Lake during 2002–2021.

**Figure 4 sensors-23-01661-f004:**
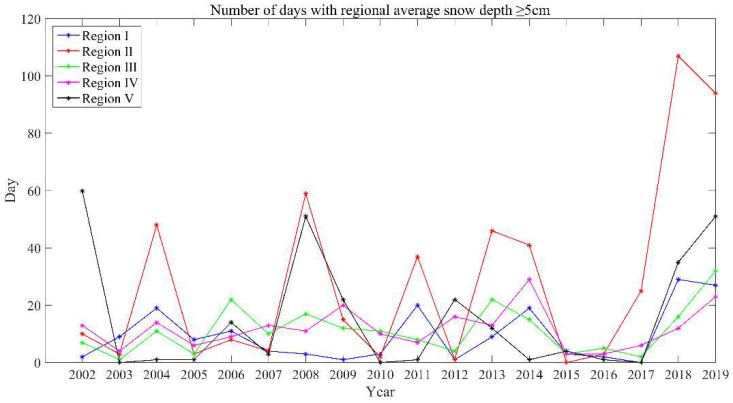
Changes in the number of days with an average snow depth ≥ 5 cm in each region from December to May of the following year. The division of regions is shown in Figure 1.

**Figure 5 sensors-23-01661-f005:**
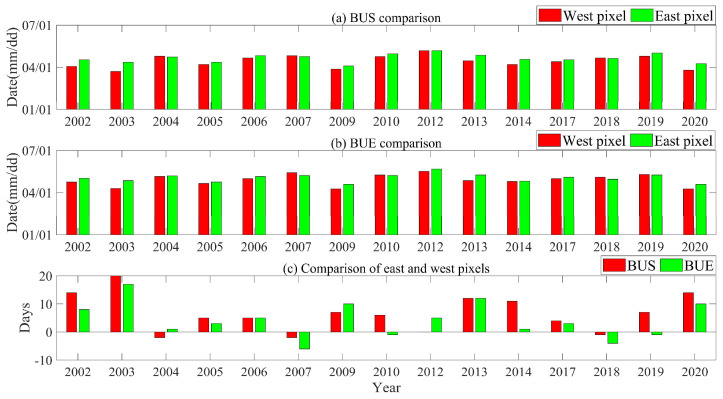
Comparison of break–up dates between east and west pixels in Nam Co during 2002 to 2021. (**a**) BUS in east and west pixels; (**b**) BUE in east and west pixels; (**c**) Difference of BUS and BUE between east and west pixels.

**Table 1 sensors-23-01661-t001:** Lake attributes and average lake ice phenology parameters. The average lake ice phenology parameters are the integration results of eight datasets (this study and seven datasets [18,30,31,32,47,50,51]). Acronyms denote start of ice break-up (BUS), end of ice break-up (BUE), start of freeze-up (FUS), end of freeze-up (FUE) dates, and ice coverage duration (ID).

FID	Name	Longitude (°)	Latitude (°)	Altitude (m)	Area (km^2^)	Averaged Ice Phenology Events				
						FUS	FUE	BUS	BUE	ID
1	Karakul Lake	73.4	39.1	3919	390.30	26 November	14 December	11 May	27 May	182
2	Ayakkum Lake	89.5	37.6	3879	727.96	11 December	26 December	5 March	20 March	100
3	Har Lake	97.6	38.3	4075	583.95	14 November	23 November	8 May	4 June	202
4	Gozha Co	81.1	35.1	5077	246.20	18 November	10 December	18 June	5 July	229
5	Bangong Co	79.8	33.6	4244	628.59	16 December	30 December	28 March	13 April	118
6	Lumajiangdong Co	81.6	34.1	4816	348.09	24 November	6 December	13 May	30 May	187
7	Aqqikkol Lake	88.4	37.1	4257	395.96	14 November	24 November	23 April	9 May	176
8	Jingyu Lake	89.4	36.4	4718	255.83	5 November	18 November	22 May	8 June	215
9	Dogaicoring Qangco	89.2	35.4	4792	292.08	8 November	17 November	30 April	9 May	183
10	Dogai Coring	89	34.6	4822	420.43	26 November	19 December	19 March	11 April	136
11	Kusai Lake	92.9	35.7	4484	248.85	13 November	28 November	6 May	18 May	187
12	Hoh Xil Lake	91.2	35.6	4886	307.52	27 October	12 November	6 June	22 June	239
13	Zhuonai Lake	91.9	35.6	4753	248.85	4 November	22 November	24 May	6 June	215
14	Ulan Ul Lake	90.4	34.9	4861	545.76	27 October	20 November	9 May	11 June	227
15	Lexiewudan Co	90.2	35.8	4870	237.50	17 November	1 December	28 April	14 May	178
16	Xijir Ulan Lake	90.3	35.3	4773	367.71	25 November	17 December	13 March	24 April	150
17	Migriggyangzham Co	90.3	33.5	4936	471.12	19 November	17 December	15 May	28 May	190
18	Donggei Cuona Lake	98.6	35.3	4086	228.55	4 December	19 December	29 April	12 May	159
19	Gyaring Lake	97.3	35	4291	486.62	7 November	26 November	31 March	22 April	166
20	Ngoring Lake	97.7	34.9	4269	607.80	21 November	10 December	22 April	6 May	166
21	Qinghai Lake	100.1	37	3191	4204.45	14 December	31 December	23 March	6 April	113
22	Langa Co	81.2	30.7	4565	253.67	20 December	11 January	19 April	9 May	140
23	Mapam Yumco	81.4	30.7	4581	408.36	11 January	24 January	17 April	30 April	110
24	Ngangla Ringco	83	31.6	4710	479.94	29 November	21 December	17 April	4 May	155
25	Taro Co	84.1	31.2	4565	478.91	16 January	22 January	19 February	18 April	92
26	Zhari Namco	85.5	31	4607	966.95	21 December	3 January	30 March	13 April	114
27	Paiku Co	85.6	28.9	4579	269.77	19 January	7 February	2 April	18 April	89
28	Dagze Co	87.5	31.9	4464	270.62	12 December	20 December	3 April	17 April	126
29	Urru Co	88	31.8	4554	342.91	28 December	12 January	20 March	22 April	115
30	Cuoe Lake	88.7	31.7	4565	255.53	25 November	18 December	13 March	7 April	133
31	Serling Co	89	31.9	4546	2201.81	17 December	2 January	3 April	18 April	122
32	Gyaring Co	88.4	31.1	4652	467.33	27 December	15 January	1 April	20 April	114
33	Ngangze Co	87.2	31.1	4682	427.97	27 November	7 December	14 March	2 April	127
34	Tangra Yumco	86.5	31	4534	833.40	14 January	23 January	6 March	16 April	92
35	Xuru Co	86.4	30.3	4720	207.19	29 January	9 February	13 March	18 April	79
36	Zige Tangco	90.9	32.1	4565	213.74	2 December	11 December	10 April	23 April	141
37	Bamco	90.6	31.3	4563	223.15	11 December	28 December	30 March	15 April	125
38	Nam Co	90.5	30.7	4725	1991.18	11 January	29 January	8 April	5 May	114
39	Puma Yumco	90.4	28.6	5012	285.63	26 December	13 January	12 April	3 May	128
40	Dorsoidong Co	89.9	33.5	4936	400.43	18 November	30 November	8 May	25 May	187

**Table 2 sensors-23-01661-t002:** Comparison of lake ice phenology datasets and lake temperature chain observations. NF is the abbreviation for not frozen.

	**Bangong Co (2012–2013)**				**Dagze Co (2012–2013)**				**Paiku Co (2016–2017)**		
	**FUS**	**FUE**	**BUS**	**BUE**	**FUS**	**FUE**	**BUS**	**BUE**	**FUS**	**FUE**	**BUE**
Cai et al., 2019 [50]					9 December	14 December	13 April	17 April			
Guo et al., 2022 [51]	12 December	12 January	22 April	2 May	11 December	19 December	3 April	16 April	6 January		15 April
Qiu, 2018 [47]					2 December	12 December	11 April	16 April			
Qiu, 2019 [31]											
This study	20 December	26 December	25 April	1 May	7 December	14 December	13 April	19 April	NF	NF	NF
Lake temperature observation	1 December	22 December	6 April	26 April	17 November	13 December	25 February	17 April			
	**Nam Co (2011–2012)**				**Nam Co (2012–2013)**				**Nam Co (2013–2014)**		
	**FUS**	**FUE**	**BUE**		**FUS**	**FUE**	**BUE**		**FUS**	**FUE**	**BUE**
Cai et al., 2019 [50]	11 January		9 May		6 January		24 May		10 January		15 May
Guo et al., 2022 [51]	4 January	28 January	9 May		7 January	27 January	12 May		4 January	5 February	10 May
Qiu, 2018 [47]	13 January	23 January	27 April		18 January	21 January	18 May		10 January	21 January	24 April
Qiu, 2019 [31]					4 January	23 January	22 May		14 January	21 January	25 April
This study					8 January	11 January	21 May		10 January	23 January	27 April
Lake temperature observation	23 December	23 January	2 May		16 December	13 January	10 May		17 December	16 January	1 May

**Table 3 sensors-23-01661-t003:** Correlation coefficients between lake ice phenology parameters and influencing factors. The black bold numbers represent the statistical significance at the 0.1 level (*t*-test).

	FUS	FUE	BUS	BUE
altitude	−0.10	−0.06	**0.28**	**0.38**
longitude	**−0.30**	**−0.29**	0.00	−0.04
latitude	**−0.71**	**−0.72**	**0.44**	**0.41**
area	0.20	0.22	−0.22	**−0.27**
average temperature in December	**0.77**	**0.75**	**−0.67**	**−0.66**
average temperature in June	**0.65**	**0.61**	**−0.74**	**−0.79**
average wind speed in December	−0.21	−0.13	0.22	**0.33**
average annual net shortwave radiation	**0.54**	**0.53**	−0.24	−0.16
average snow depth from December to May	**−0.41**	**−0.39**	**0.32**	**0.29**

**Table 4 sensors-23-01661-t004:** Mann–Kendall trend analysis results of lake ice phenological parameters from 2002 to 2021. FUS_Z, FUE_Z, BUS_Z, BUE_Z, ID_Z are the Z statistics of Mann–Kendall trend test of freeze-up start date, freeze-up end date, break-up start date, break-up end date, and lake ice cover duration, respectively. FUS_β, FUE_β, BUS_β, BUE_β, ID_β are the trend value of freeze-up start date, freeze-up end date, break-up start date, break-up end date, and lake ice cover duration, respectively. The black bold numbers represent the statistical significance at the 0.1 level (Mann–Kendall test).

FID	Name	FUS_Z	FUE_Z	BUS_Z	BUE_Z	ID_Z	FUS_β	FUE_β	BUS_β	BUE_β	ID_β
1	Karakul Lake	1.22	−0.98	1.12	1.22	0.00	0.27	−0.50	0.35	0.29	0.00
2	Ayakkum Lake	**−3.36**	**−3.71**	**4.72**	**4.69**	**4.72**	**−1.57**	**−1.40**	**3.36**	**3.25**	**5.14**
3	Har Lake	**1.96**	**1.64**	0.70	1.12	−0.42	**0.40**	**0.44**	0.43	0.50	−0.50
4	Gozha Co	**1.29**	1.06	0.23	0.34	0.00	**0.40**	0.22	0.10	0.20	0.00
5	Bangong Co	0.00	0.14	−0.59	−0.42	−0.45	0.00	0.00	−0.23	−0.17	−0.40
6	Lumajiangdong Co	0.91	0.31	−0.52	−0.07	−0.14	0.18	0.06	−0.29	0.00	−0.06
7	Aqqikkol Lake	**−2.10**	−0.73	**2.31**	**2.31**	**2.94**	**−0.33**	−0.23	**0.88**	**1.00**	**1.38**
8	Jingyu Lake	**1.54**	−0.14	**2.73**	**2.94**	**2.20**	**0.33**	0.00	**1.21**	**1.43**	**1.33**
9	Dogaicoring Qangco	**2.31**	**2.59**	**3.25**	**3.46**	**2.13**	**0.50**	**0.57**	**1.20**	**1.31**	**0.75**
11	Kusai Lake	**2.59**	**2.41**	0.84	**1.33**	−0.17	**0.50**	**0.45**	0.30	**0.41**	0.00
12	Hoh Xil Lake	**2.48**	**1.92**	**1.68**	**1.85**	0.28	**0.75**	**0.50**	**0.58**	**0.80**	0.00
13	Zhuonai Lake	**−1.33**	**−2.13**	−0.70	0.28	**1.61**	**−0.33**	**−0.33**	−0.29	0.13	**0.60**
14	Ulan Ul Lake	**4.02**	**1.57**	1.26	−0.10	**−1.61**	**0.91**	**0.38**	0.38	0.00	**−0.75**
15	Lexiewudan Co	**−3.11**	**−3.60**	**4.55**	**4.69**	**4.69**	**−1.17**	**−1.25**	**3.50**	**3.42**	**4.20**
17	Migriggyangzham Co	**−1.61**	**−1.75**	−0.38	0.07	**1.47**	**−0.50**	**−0.50**	−0.25	0.00	**0.59**
18	Donggei Cuona Lake	0.91	0.00	1.08	**1.33**	0.35	0.22	0.00	0.40	**0.45**	0.25
19	Gyaring Lake	0.31	**1.82**	0.59	0.87	0.49	0.00	**0.36**	0.43	0.67	0.33
20	Ngoring Lake	**1.36**	**1.40**	1.19	1.26	0.10	**0.50**	**0.44**	0.67	0.67	0.00
21	Qinghai Lake	**2.69**	1.08	0.00	**1.40**	−0.03	**0.50**	0.22	0.00	**0.43**	0.00
22	Langa Co	−1.01	**−1.50**	0.03	−0.21	−0.14	−0.25	**−0.50**	0.08	−0.33	−0.25
23	Mapam Yumco	0.98	0.63	−0.45	−0.21	−0.77	0.20	0.25	−0.25	−0.20	−0.63
24	Ngangla Ringco	−0.24	**−1.71**	**1.57**	**1.75**	**1.36**	−0.09	**−0.67**	**0.50**	**0.50**	**0.90**
26	Zhari Namco	−0.31	−0.77	**1.36**	0.42	0.77	−0.13	−0.17	**0.50**	0.17	0.25
28	Dagze Co	**1.40**	**1.50**	0.94	0.49	0.00	**0.33**	**0.31**	0.50	0.27	0.00
29	Urru Co	0.07	**1.75**	0.80	0.31	0.98	0.00	**0.50**	0.33	0.33	0.42
30	Cuoe Lake	0.35	−0.63	−0.35	−0.49	−0.52	0.08	−0.33	−0.13	−0.25	−0.36
31	Serling Co	0.03	−0.52	0.91	0.28	0.00	0.00	−0.19	0.33	0.15	0.00
32	Gyaring Co	1.15	0.17	0.14	0.49	−0.73	0.35	0.08	0.00	0.22	−0.33
33	Ngangze Co	0.49	0.56	1.05	0.70	0.66	0.13	0.14	0.60	0.33	0.45
36	Zige Tangco	**1.96**	**1.96**	0.21	−1.01	**−1.78**	**0.69**	**0.44**	0.00	−0.25	**−0.57**
37	Bamco	**3.08**	**1.57**	0.94	0.80	−0.70	**0.64**	**0.43**	0.43	0.33	−0.36
38	Nam Co	0.63	0.63	0.21	0.31	−0.10	0.23	0.45	0.20	0.31	−0.11
39	Puma Yumco	0.77	0.28	0.73	0.45	0.00	0.33	0.07	0.33	0.30	0.00
40	Dorsoidong Co	−0.84	−0.84	**3.43**	**3.01**	**3.15**	−0.33	−0.23	**1.80**	**1.45**	**2.08**

**Table 5 sensors-23-01661-t005:** Mann–Kendall trend analysis results of lake ice phenological parameters from 2002 to 2018. The black bold numbers represent the statistical significance at the 0.1 level (Mann–Kendall test).

FID	Name	FUS_Z	FUE_Z	BUS_Z	BUE_Z	ID_Z	FUS_β	FUE_β	BUS_β	BUE_β	ID_β
1	Karakul Lake	**1.89**	−0.32	−0.05	−0.27	−1.22	**0.67**	−0.24	0.00	−0.05	−0.61
2	Ayakkum Lake	**−3.02**	**−3.15**	**3.78**	**3.74**	**3.92**	**−1.95**	**−1.74**	**3.11**	**2.92**	**4.95**
3	Har Lake	**2.52**	**1.94**	−1.04	−0.50	**−2.39**	**0.82**	**0.57**	−0.40	−0.26	**−1.28**
4	Gozha Co	1.09	0.40	−1.19	**−1.39**	**−1.53**	0.50	0.13	−0.55	**−0.73**	**−1.20**
5	Bangong Co	1.17	0.81	−0.23	0.18	−0.68	0.33	0.30	−0.11	0.08	−0.71
6	Lumajiangdong Co	**1.49**	**1.76**	−0.77	**−1.40**	**−1.62**	**0.45**	**0.46**	−0.56	**−0.50**	**−0.96**
7	Aqqikkol Lake	**−2.43**	**−1.89**	0.99	0.86	**1.67**	**−0.50**	**−0.71**	0.50	0.35	**0.93**
8	Jingyu Lake	0.90	−0.59	**1.98**	**2.25**	**1.80**	0.17	−0.29	**0.94**	**1.32**	**1.24**
9	Dogaicoring Qangco	**2.16**	**2.25**	**2.88**	**2.97**	**1.35**	**0.75**	**0.50**	**1.11**	**1.37**	**0.58**
11	Kusai Lake	**2.84**	**2.34**	−0.59	0.00	**−1.98**	**0.60**	**0.45**	−0.25	0.00	**−0.64**
12	Hoh Xil Lake	**3.42**	**2.03**	0.95	**1.58**	−0.41	**1.29**	**0.78**	0.44	**0.68**	−0.24
13	Zhuonai Lake	**−1.76**	**−2.79**	**−2.48**	−1.17	0.23	**−0.44**	**−0.63**	**−0.79**	−0.33	0.06
14	Ulan Ul Lake	**3.38**	0.50	0.23	−0.81	**−1.94**	**1.00**	0.08	0.10	−0.48	**−1.33**
15	Lexiewudan Co	**−2.61**	**−2.93**	**3.83**	**3.92**	**3.92**	**−1.19**	**−1.36**	**3.41**	**3.20**	**4.00**
17	Migriggyangzham Co	**−1.49**	**−1.31**	**−1.44**	−0.77	0.41	**−0.57**	**−0.50**	**−1.00**	−0.60	0.21
18	Donggei Cuona Lake	**1.44**	0.27	0.36	0.81	−0.45	**0.41**	0.10	0.26	0.39	−0.25
19	Gyaring Lake	−0.27	**1.44**	−0.09	0.32	0.05	0.00	**0.28**	−0.13	0.35	0.10
20	Ngoring Lake	1.04	0.59	0.32	0.23	−0.63	0.50	0.32	0.33	0.33	−0.38
21	Qinghai Lake	**2.12**	0.99	−0.86	0.00	−1.26	**0.60**	0.37	−0.50	0.00	−0.85
22	Langa Co	−0.27	−0.63	0.00	−0.59	−0.50	−0.24	−0.32	−0.08	−0.55	−0.63
23	Mapam Yumco	**1.49**	**1.71**	−0.14	−0.27	−1.13	**0.58**	**1.00**	−0.11	−0.27	−1.56
24	Ngangla Ringco	0.77	−0.09	0.77	0.86	0.14	0.33	0.00	0.19	0.32	0.00
26	Zhari Namco	0.36	0.09	0.41	−0.14	−0.18	0.32	0.00	0.20	−0.12	0.00
28	Dagze Co	**2.07**	**1.94**	0.63	0.09	−0.81	**0.74**	**0.68**	0.29	0.00	−0.42
29	Urru Co	0.90	**2.25**	0.41	0.14	0.54	0.50	**0.68**	0.31	0.25	0.33
30	Cuoe Lake	0.72	0.18	−0.77	**−1.62**	**−1.67**	0.17	0.03	−0.54	**−1.00**	**−0.83**
31	Serling Co	0.77	0.45	0.59	−0.23	−0.90	0.52	0.11	0.33	−0.09	−0.67
32	Gyaring Co	**2.48**	1.26	−0.09	−0.09	**−1.89**	**0.84**	0.68	0.00	−0.05	**−0.87**
33	Ngangze Co	0.95	**1.71**	1.13	0.95	0.77	0.33	**0.78**	0.91	0.52	0.57
36	Zige Tangco	**2.48**	**2.21**	−0.45	**−1.49**	**−2.48**	**1.15**	**1.00**	−0.13	**−0.37**	**−1.10**
37	Bamco	**3.56**	**1.89**	0.54	0.18	**−1.53**	**1.16**	**1.00**	0.35	0.20	**−1.27**
38	Nam Co	0.77	0.95	0.36	0.27	−0.54	0.33	1.00	0.44	0.50	−0.73
39	Puma Yumco	1.22	0.05	0.18	0.23	−0.27	0.41	0.00	0.07	0.31	−0.12
40	Dorsoidong Co	**−1.62**	−1.17	**2.66**	**2.12**	**2.52**	**−0.58**	−0.40	**1.46**	**1.21**	**1.96**

**Table 6 sensors-23-01661-t006:** Comparison of trend analysis results between 2002–2018 and 2002–2021. The number (proportion) of lakes with earlier/later FUS, BUE, and with shorter/longer ID are determined. Their average date or days and change rate are also shown.

**Changes of Lake Ice Phenology in 34 Completely Frozen Lakes from 2002 to 2018**			
**Parameter**	**Average Date/Days**	**Number (Proportion)**	**Change Rate (Days/Year)**
FUS	22 December	26 (76.5%)	0.59
		8 (23.5%)	−0.68
BUE	22 May	18 (52.9%)	0.79
		14 (41.2%)	−0.38
ID	153	22 (64.7%)	−0.76
		12 (35.3%)	1.24
**Changes of Lake Ice Phenology in 34 Completely Frozen Lakes from 2002 to 2021**			
**Parameter**	**Average Date/Days**	**Number (Proportion)**	**Change Rate (Days/Year)**
FUS	9 December	24 (70.6%)	0.35
		9 (26.5%)	−0.52
BUE	5 May	27(79.4%)	0.72
		7 (20.6%)	−0.17
ID	147	13 (38.2%)	−0.33
		16(47.1%)	1.17

## Data Availability

Not applicable.

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
