# Peer review of "Analysis of Ice Phenology of Middle and Large Lakes on the Tibetan Plateau"

_sensors, 2023, doi:10.3390/s23031661_

Round 1

Reviewer 1 Report

Authors must emphasize the value of observing ice from space and its significance as a good indicator of climate change in the introduction section.

When comparing optical and microwave remote sensing, authors must consider the advantages and disadvantages of each technique.

Authors also require addressing what is the significant of e the lake ice phenology investigation of the Tibetan Plateau.

In the introduction section, authors are required to address advanced image processing algorithm such as genetic algorithm for ice phenology detection, especially in microwave remote sensing data. I suggest the following reference to be added:

Marghany M. Introductory Chapter: Automatic Detection of Ice Covers in Airborne Radar Data Using Genetic Algorithm. In Recent Remote Sensing Sensor Applications-Satellites and Unmanned Aerial Vehicles (UAVs) 2022 Oct 26. Intech Open.

It does not address probably the novelty of this work.

How were the brightness temperature filters created by authors? This assertion is ambiguous. Another equation serving as an index in the fluctuations of the ice phenology parameters is required to correlate with equations 1 and 2.

Therefore, it is not clear how curves Y1 and Y4 are used to determine 204 FUS, and curves Y2 and Y3 are used to determine BUE. Equations 3 to 6 are not clear! The authors are required to establish such a boundary conditions criteria to determine the dynamic fluctuations of the ice phenology as a function of brightness temperature. 

How to correlate logically equation 7 by equations 1 to 6?

Authors are required to show the output of remote sensing data is used with implementations of equations 1 to 6. In addition, authors are required to show advanced approach to get the accuracy of this work using the receiver operating characteristic (ROC) technique. 

I cannot accept the manuscript for publication until all the aforementioned comments are solved.

Author Response

Dear Reviewer:

Please see the attachment and please find the reversion of sensors-2136083 “Analysis of ice phenology of middle and large lakes on the Tibetan Plateau” by Lijun Sun, Binbin Wang, Yaoming Ma, Xingdong Shi, Yan Wang.

We really appreciate the constructive comments from you to improve our revision. We have revised the manuscript accordingly. The detailed, point-by-point response to the review comments in the Revision Note file(author-coverletter-25781150.v2.pdf).

Reviewer 2 Report

The remote sensing has been widely used for ice phenology monitoring because of its high temporal and spatial resolutions. The application of different remote sensing data and methods has filled and expanded lake ice monitoring records, and the results of lake ice phenology trend analysis reveal a general phenomenon that under the background of global warming, most of the lakes will freeze up later, break up earlier, and have shorter ice coverage duration. This manuscript used AMSR-E, AMSR-2, MODIS data and in situ measurements of temperature profiles in 4 lakes, ice-phenology events of 40 high-elevation large lakes are derived and their inter-annual trends and influencing factors are analyzed. The results, conclusions and suggestions are useful to technicians who use remote sensing for analysis the ice phenology of lake. However, the authors could do much better job on analyzing and presenting the data before publishing. Followings suggestions still need to be improved:

1)      Under climate warming, ice coverage duration (ID) in most lakes should be reduced. The results show that only 13 (38.2%) lakes shortened ID at an average rate of -0.33 days/year, but there are 16 (47.1%) lakes extended ID at an average rate of 1.17 days/year. What caused this?

2)      The use of variable symbols in the formula is not standardized, and the author needs to verify one by one, such as, ’TBi-1’ symbol expression is incorrect in formula (1) ~ (6).

3)      L328: the average lake ice duration is meaningless, because the freezing and thawing dates of lakes vary greatly, mainly due to different meteorological conditions and local factors.

4)      It is recommended to delete Figure 3 (b).

5)      L501: the trend towards later FUS is still common (76.5% of the lakes, average change rate = 0.59 days/year) is meaningless.

Author Response

Please see the attachment and please find the reversion of sensors-2136083 “Analysis of ice phenology of middle and large lakes on the Tibetan Plateau” by Lijun Sun, Binbin Wang, Yaoming Ma, Xingdong Shi, Yan Wang.

We really appreciate the constructive comments from you to improve our revision. We have revised the manuscript accordingly. The detailed, point-by-point response to the review comments in the Revision Note file (author-coverletter-26363726.v1.pdf).

Round 2

Reviewer 1 Report

The authors have done excellent revision. So I highly recommend it for publication. This work is innovative and well-researched, addressing questions and themes not previously explored .